# Superiority of Direct Oral Anticoagulants over Vitamin K Antagonists in Oncological Patients with Atrial Fibrillation: Analysis of Efficacy and Safety Outcomes

**DOI:** 10.3390/jcm11195712

**Published:** 2022-09-27

**Authors:** Iris Parrini, Fabiana Lucà, Carmelo Massimiliano Rao, Gianmarco Parise, Linda Renata Micali, Giuseppe Musumeci, Mark La Meir, Furio Colivicchi, Michele Massimo Gulizia, Sandro Gelsomino

**Affiliations:** 1Division of Cardiology, Mauriziano Hospital, 10128 Turin, Italy; 2Grande Ospedale Metropolitano, 89124 Reggio Calabria, Italy; 3Department of Cardiothoracic Surgery, Cardiovascular Research Institute, Maastricht University, 6211 LK Maastricht, The Netherlands; 4University Hospital Brussels, 1050 Brussels, Belgium; 5Cardiology Unit, San Filippo Neri Hospital, 00135 Roma, Italy; 6Cardiology Division, Ospedale Garibaldi-Nesima, Azienda di Rilievo Nazionale e Alta Specializzazione “Garibaldi”, 95126 Catania, Italy

**Keywords:** direct oral anticoagulants, warfarin, safety, efficacy, atrial fibrillation, cancer

## Abstract

**Background and aim**. Cancer and atrial fibrillation (AF) may be associated, and anticoagulation, either with vitamin K antagonists (VKAs) or direct oral anticoagulants (DOACs), is necessary to prevent thromboembolic events by reducing the risk of bleeding. The log incidence rate ratio (IRR) and 95% confidence interval were used as index statistics. Higgin’s I^2^ test was adopted to assess statistical inconsistencies by considering interstudy variations, defined by values ranging from 0 to 100%. I^2^ values of less than 40% are associated with very low heterogeneity among the studies; values between 40% and 75% indicate moderate heterogeneity, and those greater than 75% suggest severe heterogeneity. The aim of this meta-analysis was to compare the safety and efficacy of VKAs and DOACs in oncologic patients with AF. **Methods**. A meta-analysis was conducted comparing VKAs to DOACs in terms of thromboembolic events and bleeding. A meta-regression was conducted to investigate the differences in efficacy and safety between four different DOACs. Moreover, a sub-analysis on active-cancer-only patients was conducted. **Results**. A total of eight papers were included. The log incidence rate ratio (IRR) for thromboembolic events between the two groups was −0.69 (*p* < 0.005). The meta-regression did not reveal significant differences between the types of DOACs (*p* > 0.9). The Log IRR was −0.38 (*p* = 0.008) for ischemic stroke, −0.43 (*p* = 0.02) for myocardial infarction, −0.39 (*p* = 0.45) for arterial embolism, and −1.04 (*p* = 0.003) for venous thromboembolism. The log IRR for bleeding events was −0.43 (*p* < 0.005), and the meta-regression revealed no statistical difference (*p* = 0.7). The log IRR of hemorrhagic stroke, major bleeding, and clinically relevant non-major bleeding between the VKA and DOAC groups was −0.51 (*p* < 0.0001), −0.45 (*p* = 0.03), and 0.0045 (*p* = 0.97), respectively. Similar results were found in active-cancer patients for all the endpoints except for clinically-relevant non-major bleedings. **Conclusions**. DOACs showed better efficacy and safety outcomes than VKAs. No difference was found between types of DOACs.

## 1. Introduction 

The association between cancer and atrial fibrillation (AF) has been well assessed [1]. The relationship between these two entities might be bi-directional. On one hand, cancer patients are more likely to develop AF; on the other hand, patients with AF should be accurately investigated in order to rule out unknown cancer [1,2,3]. A lot of underlying mechanisms have been hypothesized. Firstly, cancer-derived systemic inflammation has been shown to promote atrial remodeling, and autonomous nervous system dysregulation might increase the risk of developing AF [2]. Moreover, cancer treatments as radiotherapy and chemotherapeutic agents have been associated with cardiotoxic effects [2,4]. Furthermore, oncological surgery may cause the onset of AF [5]. Moreover, it is noteworthy that cancer patients with AF have not only the AF-related thromboembolic risk but also an intrinsic risk due to hypercoagulability and the prothrombotic state typically occurring in malignancy. It has been well assessed that tumor cells promote the activation of the coagulation system, secreting procoagulant factors and inflammatory cytokines. A complex interaction between tumor cells and blood and vascular cells has also been reported [6,7].

Therefore, in cancer patients with AF, an anticoagulation strategy is needed to be adopted to face the risk of thromboembolic events related to AF [7,8]. Treatment with VKAs and or low-molecular-weight heparin (LMWH) has been traditionally considered [9,10,11] as the elective approach; however, the introduction of DOACs has raised matters on the efficacy and safety of the types of anticoagulation in patients with cancer and AF [12]. Notably, in patients with cancer and VTE, DOACs demonstrated equal or superior benefits compared to VKAs and LMWH [13]. Conversely, similar data regarding the use of DOACs for the prevention of thromboembolism in AF are lacking [14] 

Nevertheless, this has been largely excluded from trials. Therefore, we conducted a meta-analysis of these studies with the aim of assessing the safety and efficacy of VKAs and DOACs in oncologic patients concomitantly affected by AF. 

## 2. Methods 

### 2.1. Search Strategy

A literature search was performed according to the principles of the Preferred Reporting Items for Systematic Review and Metanalyses (PRISMA) [15] register on Prospero, with the number: CRD42022348866. The search strategy was determined by two authors (F.L. and L.M.) and approved by another reviewer (I.P.). Reference lists of the papers obtained through the literature search were screened in order to possibly include a larger number of relevant studies. Titles and abstracts of all articles published in the last 15 years were initially assessed.

The literature search was performed by one investigator (L.M.) and was focused on the identification of articles examining anticoagulant therapy (either VKA or DOAC) in patients with the coexistence of cancer and AF. The search engines selected for this meta-analysis were PubMed and EMBASE Databases. The search strategy included the following search Boolean and Mesh terms: “Neoplasms” [Mesh] AND “Atrial Fibrillation” [Mesh] AND “Warfarin” [Mesh] AND “Anticoagulants” [Mesh], “Cancer” OR “Neoplasm” AND “Atrial Fibrillation” AND “Warfarin” AND “Direct Oral Anticoagulant”. 

### 2.2. Selection Process

Articles selection was based on the following inclusion criteria: (1) studies reporting a comparison between VKAs and DOACs in terms of efficacy and safety; (2) studies including patients with either active cancer (newly diagnosed or actively treated) or remote cancer (history of cancer); (3) studies with cohorts of more than 10 patients; and (4) human studies. The exclusion criteria were: (1) non-comparative studies; (2) lack of usable data concerning efficacy and safety of VKAs and DOACs; (3) studies reporting mixed data for patients affected by either venous thromboembolism or atrial fibrillation; (4) studies with patient cohorts comprising 10 or fewer individuals. 

### 2.3. Quality Assessment 

The quality of included studies was assessed using Down and Black’s Checklist for Measuring, which evaluates the quality of randomized and non-randomized studies in terms of reporting, external validity, internal validity, and power [16]. Each component of the checklist is rated using a binary score (0/1) except for two items which are rated on a scale from 0 to 2 and from 0 to 5, respectively [16]. Two independent researchers (L.M. and G.P) conducted the ratings. Divergences were resolved by quantification through Cohen’s kappa [17]. 

### 2.4. Endpoints and Definitions 

The analysis of the endpoints was conducted in patients with remote 156 and active cancer with an additional sub-analysis of only patients with 157 active cancer. The primary endpoints were the safety and efficacy of DOACs vs. VKAs. Safety as a composite outcome included major bleeding, hemorrhagic stroke, clinically relevant non-major bleeding (CRNMB) and minor bleeding [18,19]. Efficacy was defined as either composite or non-composite outcomes and included venous thromboembolism, arterial thromboembolism, ischemic stroke, and myocardial infarction [20,21,22]. 

The analysis of the endpoints was conducted in patients with remote and active cancer, with an additional sub-analysis of only patients with active cancer. 

### 2.5. Statistical Analysis 

Values were expressed as mean ±S Standard Deviation (SD) or median interquartile range in case of normal or non-normal distribution, respectively. A meta-analysis was conducted using v. 3.6.1 (R Foundation for Statistical Computing, Vienna, Austria). The log incidence rate ratio (IRR) and 95% confidence interval [23] were used as index statistics. The IRR was used since the follow-up periods were different [24]. The log transformation makes the outcome measure symmetric at 0, yielding the sampling distribution closer to normality.

Higgin’s I^2^ test was adopted to assess statistical inconsistency, by considering interstudy variation, defined by values ranging from 0 to 100%. Values of I^2^ less than 40% are associated with very low heterogeneity among studies, values between 40% and 75% indicate moderate heterogeneity, and those greater than 75% suggest severe heterogeneity [25]. Since a high degree of heterogeneity between studies was expected, a random effects model was used. Publication bias was evaluated using Egger’s test for the intercept. In addition, a meta-regression analysis was performed with the aim of investigating the impact of each class of DOAC on survival. *p* values < 0.05 were considered statistically significant.

Additionally, because of the prevalence of one study over the others (Sawant et al. [26]), a sensitivity analysis was performed, excluding this study. 

## 3. Results

### 3.1. Characteristics of Studies and Population 

Figure 1 shows the schematic representation of the selection process. At the end of this process, eight papers were retained [19,27,28,29,30,31,32]. The papers retrieved were published between 2017 and 2020. A total of four papers were prospective randomized studies [19,27,28,29], three were retrospective cohort studies [30,31,32], and one was a retrospective observational study [26]. The total number of patients included was 228,497 (range 224–196,517). These patients had either active cancer or remote/history of cancer. The definitions of active and remote cancer attributed by each paper are reported in Table 1. A total of six papers included patients with active cancer [19,20,21,22,23,24,25,26,29,30,31,32], whereas the other two included patients with either active cancer or remote cancer [27,28]. Overall, 226,828 (99.3%) patients had active cancer and were treated, and 1669 (0.7%) had remote cancer. In particular, in the VKA group, 181,232 (99.5%) patients had active cancer and 844 (0.5%) had remote cancer; in the DOAC group, 45,596 (98.2%) subjects had active cancer, and 825 (1.8%) had remote cancer. In our population, 182,076 (79.7%) were treated with VKA, which was warfarin in most cases, whereas 46,421 (20.3%) received a DOAC. Seven studies specified the type of DOAC [19,26,27,28,30,31,32]. Thus, our analysis included 11,372 (24.5%) patients treated with apixaban, 15,148 (32.6%) with rivaroxaban, 17,322 (37.3%) with dabigatran, and 770 (1.7%) with edoxaban (Table 1). It is unknown whether the remaining 1809 (3.9%) patients received apixaban, rivaroxaban, dabigatran, or edoxaban. The dosages of DOACs and VKAs used in each study are shown in Appendix A. 

The mean age was 74.6 [73.8–75.5] in the overall population, 74.3 [72.3–76.4] in the VKA group, and 74.1 [72.8–75.5] in the DOAC group. The majority of papers reported the CHA_2_DS_2_ VASc score and the HAS-BLED score for the assessment of the thrombotic and hemorrhagic risks, respectively. A total of four papers reported the overall CHA_2_DS_2_ VASc mean score and the HAS-BLED mean score [19,27,28,31], while three studies reported the CHA_2_DS_2_ VASc mean score and the HAS-BLED mean score for both the VKA group and the DOAC group [30,31,32] (Appendix A). Malignancy characteristics of the included patients are shown in Appendix A. 

### 3.2. Quality of the Studies 

The average overall quality rating was 0.86 ± 0.42, with ratings ranging from 0.38 to 1.56. Appendix A presents the average scores for the items on the checklist. The analysis revealed lower scores related to the internal validity for both bias and selection bias and for power analysis, which is related to the quality of reporting. Acceptable interrater agreement was found (κ = 0.61; %-agree = 84.3).

### 3.3. Follow Up 

The follow up was 100% completed in seven studies [19,26,28,29,30,31,32]. Therefore, 227,857 (99.7%) patients reached the end of the follow-up period. The follow-up period ranged from 1–4 years. 

### 3.4. Safety Outcomes

Table 2 reports the safety outcomes from the follow-up. All papers reported data concerning the safety of VKAs versus DOACs. As shown in Figure 2A, the log IRR of bleeding events between VKAs and DOACs was −0.43 [95% CI: −0.66, −0.20] (*p* = 0.0002; I^2^ = 89.15%, *p* < 0.0001; Egger’s test: intercept −0.19 [95% CI: −0.54, 0.17], *p* = 0.33). This suggests that the DOACs are superior to VKAs in protecting patients from bleeding events, in terms of reducing the risk of major bleeding, confirming previous data [33]. Appendix A shows the funnel plot for bias. In contrast, as shown in Figure 2B, the regression analysis revealed no statistical difference in terms of bleeding in relation to the type of DOAC (*p* = 0.7). Statistically significant results were found in the sub-analysis on patients with only active cancer, with DOACs being superior to VKAs (log IRR: −0.49 [95% CI: −0.71, −0.26], *p* < 0.0001; I^2^ = 83.58%, *p* < 0.0002; Egger’s test: intercept −0.31 [95% CI: −0.62, 0.00], *p* = 0.003; forest plot in Appendix A; funnel plot in Appendix A). The analysis without Sawant et al. [26] on safety confirmed significant results in favor of DOACs both in remote + active-cancer patients (forest plot in Appendix A; funnel plot in Appendix A) and in active-cancer-only patients (forest plot in Appendix A; funnel plot in Appendix A). 

### 3.5. Efficacy Outcomes

Table 2 reports the efficacy outcomes at follow-up. The results of the analysis without Sawant et al., both in remote + active-cancer patients and active-cancer-only patients are presented in Appendix A. All papers reported data concerning the efficacy of VKAs versus DOACs. As shown in Figure 3A, the log IRR of the total thromboembolic events between VKA and DOAC was −0.6921 [95% CI: −1.1097, −0.2745] (*p* = 0.0012; I^2^ = 96.20%, *p* < 0.0001; Egger’s test: intercept −0.06 [95% CI: −0.18, 0.06], *p* = 0.34), showing that DOACs are superior in preventing the occurrence of thromboembolic events when compared to VKAs. Appendix A shows the funnel plot for bias. The meta-regression did not reveal statistically significant differences between types of DOACs in terms of thromboembolic events (*p* > 0.9) (Figure 3B). In patients with active cancer, thromboembolic events were more likely to occur in the VKA group rather than the DOAC group (log IRR: −0.83 [95% CI: −1.25, −0.40], *p* = 0.0001; I^2^ = 95.78%, *p* < 0.0001; Egger’s test: intercept −0.05 [95% CI: −0.16, 0.06], *p* = 0.01; forest plot in Appendix A; funnel plot in Appendix A). The analysis without Sawant et al. [26] on efficacy confirmed the significant results in favor of DOACs both in remote + active-cancer patients (forest plot in Appendix A; funnel plot in Appendix A) and in active-cancer-only patients (forest plot in Appendix A; funnel plot in Appendix A). The occurrence of ischemic stroke was reported in all the studies included, as visible in Figure 3C. The log IRR between VKAs and DOACs (−0.38 [95% CI: −0.66, −0.10]) revealed a significant statistical difference between the two groups (*p* = 0.008), with the VKAs being associated with a higher incidence (I^2^ = 68.67%, *p* = 0.004; Egger’s test: intercept −0.09 [95% CI: −0.14, −0.04], *p* = 0.002; funnel plot in Appendix A). In the sub-analysis on active cancer only, the VKA group showed higher incidence of ischemic stroke compared to the DOAC group (log IRR: −0.42 [95% CI: −0.69, −0.14], *p* = 0.04; I^2^ = 65.76%, *p* = 0.01; Egger’s test: intercept −0.08 [95% CI: −0.12, 0.04], *p* = 0.00; forest plot in Appendix A; funnel plot in Appendix A). The analysis without Sawant et al. [26]. Table 3 on ischemic stroke confirmed the significant results in favor of DOACs both in remote + active-cancer patients (forest plot in Appendix A; funnel plot in Appendix A) and in active-cancer-only patients (forest plot in Appendix A; funnel plot in Appendix A). 

Figure 3D shows that myocardial infarction was reported by four studies [19,27,29,32]. The log IRR deriving from the analysis from these papers is −0.43 [95% CI: −0.81, −0.06], suggesting that DOACs showed a significant reduction in myocardial infarction compared to VKAs (*p* = 0.02; I^2^ = 7.70%, *p* = 0.40; Egger’s test: intercept −1.75 [95% CI: −2.73, −0.78], *p* = 0.07; funnel plot in Appendix A). Similarly, in active-cancer-only patients, DOACs had a significantly lower incidence of myocardial infarction (log IRR: −0.65 [95% CI: −1.07, −0.22], *p* = 0.003; I^2^ = 0.00%, *p* = 0.8; Egger’s test: intercept −0.59 [95% CI: −1.18, 0.01], *p* = 0.21; forest plot in Appendix A; funnel plot in Appendix A).

Arterial embolism was reported by five studies [19,27,29,31,32], and the log IRR was −0.39 [95% CI: −1.38, 0.61]) derived from Figure 3E, showing no significant difference between the VKAs and DOACs (*p* = 0.45; I^2^ = 0.00%, *p* = 0.39; Egger’s test: intercept 1.27 [95% CI: −1.84, 4.39], *p* = 0.49; funnel plot in Appendix A). No significant difference was also found in the sub-analysis for active cancer only (log IRR: −0.76 [95% CI: −1.86, 0.35], *p* = 0.12; I^2^ = 0.21%, *p* = 0.6; Egger’s test: intercept 1.14 [95% CI: −0.09, 2.38], *p* = 0.08; forest plot in Appendix A; funnel plot in Appendix A).

As shown in Figure 3F, the log IRR in four studies on VTE was evaluated [27,28,29,30] and was −1.04 [95% CI: −1.71, −0.36], suggesting that DOACs are associated with a lower incidence of VTE (*p* = 0.003; I^2^ = 91.12%, *p* < 0.0001; Egger’s test: intercept −0.78 [95% CI: −1.54, −0.02], *p* = 0.04; funnel plot in Appendix A). Similarly, in active-cancer-only patients, DOACs showed a lower occurrence of VTE (log IRR: −1.29 [95% CI: −2.00, −0.57], *p* = 0.0004; I^2^ = 91.61%, *p* < 0.0001; Egger’s test: intercept −0.61 [95% CI: −1.39, 0.17], *p* = 0.05; forest plot in Appendix A; funnel plot in Appendix A). 

Hemorrhagic stroke was considered in only three of the included studies (Figure 4A). The log IRR between VKAs and DOACs was −0.51 [95% CI: −0.64, −0.38], suggesting that a significant statistical difference was found between the two anticoagulant classes (*p* < 0.0001; I^2^ = 0.00%, *p* = 0.9; Egger’s test: intercept −0.50 [95% CI: −0.57, −0.42], *p* = 0.05; funnel plot in Appendix A). DOACs showed a significantly lower incidence of hemorrhagic stroke compared to VKAs. The same result was found in active-cancer-only patients with (log IRR: −0.51 [95% CI: −0.64, −0.38], *p* < 0.0001; I^2^ = 0.00%, *p* = 0.09; Egger’s test: intercept −0.52 [95% CI: NA, NA], *p* = NA; forest plot in Appendix A; funnel plot in Appendix A). On the contrary, in the analysis without Sawant et al. [26], we found no difference between DOACs and VKAs in terms of hemorrhagic stroke (forest plot in Appendix A; funnel plot in Appendix A). Furthermore, the analysis without Sawant et al. [26] and on active-cancer-only patients could not be performed as only one paper, with the exclusion of Sawant et al., focused on active-cancer patients (Ording et al. [29]). 

As reported in Figure 4B, the log IRR for major bleeding was −0.45 [95% CI: −0.75, −0.16] (*p* = 0.03; I^2^ = 69.89%, *p* = 0.02; Egger’s test: intercept 0.06 [95% CI: −0.38, 0.49], *p* = 0.02; suggesting that major bleedings were more likely to occur in the VKA group, funnel plot in Appendix A). A similar result was found in patients with active cancer (log IRR: −0.59 [95% CI: −1.02, −0.16], *p* = 0.008; I^2^ = 80.99%, *p* = 0.006; Egger’s test: intercept 0.03 [95% CI: −0.36, 0.42], *p* = 0.03; forest plot in Appendix A; funnel plot in Appendix A). 

As shown in Figure 4C, CRNMB were solely reported in the three randomized trials included [19,27,28], revealing no statistical difference between VKAs and DOACs; funnel plot in Appendix A). Conversely, in patients with active cancer, CRNMB were more likely to occur in the VKA group rather than the DOAC group (−0.59 [95% CI: −1.02, −0.16] *p* = 0.008; I^2^ = 80.99%, *p* = 0.006; Egger’s test: intercept 0.03 [95% CI: −0.36, 0.42], *p* = 0.03; forest plot in Appendix A; funnel plot in Appendix A).

As shown, CRNMB were solely reported in the three randomized trials included [16,21,23]. 

## 4. Discussion 

The presence of AF may be a marker of occult cancer and new-onset atrial fibrillation (AF) concomitantly with cancer and may be an index of an advanced stage of the disease [2,3]. In the initial follow-up of AF, particularly in the first 3 months, cancer is more likely to be diagnosed due to heightened medical surveillance, which might reveal the presence of the time of the AF diagnosis rather than as a consequence of AF [2]. 

The diagnostic work-up for AF or its comorbidities, including clinical examinations and screening for underlying diseases, might lead to cancer detection. AF per se predisposed to thromboembolic events, with certain types of malignancies, inducing the onset of Trousseau’s syndrome [34,35]. This procoagulant state is promoted via several mechanisms, in which there is abnormal activation of the procoagulant molecules and interaction with adhesion molecules, leading to thrombus formation [36]. An increase in fibrinogen levels, platelet counts, fibrin degradation factors (e.g., D-dimer), and factors such as factor VIII frequently occur in cancer patients [6,37].

These coagulative alterations are likely to be caused not only by the activity of tumor cells but can also be induced by anti-cancer therapies and lead to an increased susceptibility to thrombosis [38]. Tumor cells are able to directly activate the coagulation cascade by secreting cells characterized by a procoagulant activity, including tissue factors, which trigger the coagulation cascade by producing a complex with factor VII [39,40]. It has been reported that tissue factor X and circulating microparticles could also be activated, while heparanase is likely to interact with the physiological inhibitor of the tissue factors, increasing tissue factor activity [41]. In addition, tumor cells are also involved in activating both other blood cells (including endothelial cells) and inflammatory cytokines and pro-angiogenic factors, which interfere with the coagulation cascade. The resulting alterations lead to a prothrombotic state [42].

Conversely, some malignancies and anticancer therapies have been shown to be associated with hemorrhagic events [3,28]. Remarkably, hematological malignancies are more likely to cause thrombocytopenia. Therefore, if an anticoagulant strategy AF is needed, the bleeding risk might rise. In this regard, a higher incidence of bleeding in oncologic patients when compared to non-oncologic individuals [19,27,28] has been reported [21,28], making the choice for the most proper AF treatment challenging [28].

Considering the scarcity of studies on the use of VKAs versus DOACs in cancer patients with AF, the population number explored in this meta-analysis may be considered quite conspicuous. The main finding of this meta-analysis is the superiority of DOACs over VKAs in reducing the incidence of both thromboembolic and hemorrhagic events in active and remote cancer. Our results are in accordance with previous meta-analyses comparing DOACs and VKAs in terms of safety and efficacy [9], even though the present meta-analysis covers a higher number of studies. Specifically, we have found a statistically significant decrease in ischemic stroke, myocardial infarction, and VTE in the DOAC group compared to the VKA group. Our results diverge from another prior meta-analysis, Yang et al. [43], because, despite a difference between the two groups in terms of stroke, arterial embolism, and VTE being reported, statistical significance has not been observed. Our efficacy outcome is noteworthy because our overall population had moderate/high thrombotic risk, as evidenced by the CHADS2 and CHA2DS2-VASc scores [44], remarking the superiority of DOACs over VKAs in decreasing the incidence of thromboembolic events. 

Furthermore, we found a difference between VKAs and DOACs in terms of major bleeding but not CRNMB. Our meta-analysis is the first to find an increased incidence of major bleeding at follow-up in the VKA group since the other meta-analyses described no difference in safety outcomes between VKAs and DOACs [9,43,45]. We found that DOACs showed better outcomes with regard to major bleeding in conditions of comparable bleeding risks across the two classes of anticoagulants. Indeed, in the included studies, which reported the HAS-BLED score [19,27,28,31], the bleeding risk was moderate (HAS-BLED < 3) [44] in both the VKA and DOAC groups. The similarities in baseline characteristics between our two groups also derive from the studies by Shah et al. [30] and Kim et al. [32], who used a propensity score with balanced characteristics, thus, strengthening, in turn, our result. Moreover, we also found that DOACs are more effective in reducing the incidence of hemorrhagic stroke. However, by conducting the analysis excluding Sawant et al. [26], we found that the difference between DOACs and VKAs in terms of hemorrhagic stroke was not statistically significant. Notably, a significant increase in hemorrhagic stroke among patients treated with VKAs has been reported by Sawant et al. [26]; conversely, although a difference between DOACs and VKAs in favor of DOACs has also been reported by other authors [27,29], it was not significative.

From our findings, DOACs performed better in patients of comparable age and with active cancer despite the fact that the VKA group included a greater number of patients. Such a result could be attributable to several factors that could of reduced the effectiveness of VKAs, such as drug–drug interactions, nutritional hindrance, difficulty in achieving an effective therapeutic response due to a narrow therapeutic index, and impaired medication efficacy in cancer patients [11]. Furthermore, in cancer patients, the control of INR can be unreliable, thus, compromising the accuracy of such a parameter [46]. Especially in oncologic patients, INR is subject to frequent fluctuations because of the interaction between warfarin and chemotherapy agents that alter VKA metabolism [46]. Another noteworthy factor that could have influenced our results is the adherence to anticoagulation treatment. It is a known fact that patients treated with VKAs tend to be less persistent in the assumption of medications when compared to patients treated with DOACs. Lack of persistence has been related to the fact that warfarin requires constant INR monitoring, which could discourage patients from using it as a long-term therapy, even in those who assume VKAs in secondary prevention [32,43]. Additionally, the irregular dietary intake of vitamin K due to anorexia, nausea or vomiting, low body weight, and low albumin can cause instability for INR. Consequently, this can lead to an over- or under-estimation of the thrombotic and bleeding risks, making these patients more likely to develop adverse events. Antithrombotic therapy might also have detrimental effects on cancer progression.

Furthermore, a detrimental effect of DOACS on cancer progression might be supposed, as it plays a role of the inhibition of factor X by apixaban, rivaroxaban, and edoxaban and factor II by dabigatran in the progression of the malignant process hypothesized [23,44,45]. In this regard, factors, X and II seem to be involved in the promotion of neo-angiogenesis and the formation of metastasis [23,45].

Noteworthily, our meta-analysis shows no difference in terms of efficacy and safety between apixaban, rivaroxaban, dabigatran, and edoxaban. In the ENGAGE AF-TIMI 48 Trial, Fanola et al. [19] found that edoxaban could be associated with better outcomes. Edoxaban seems to have lower inhibitory effects on CYP3A4 than apixaban, rivaroxaban, and dabigatran, thus, exerting minor consequences on the metabolism of many anticancer medications [18,19]. This inconsistency could be explained by the fact that only one of the papers included in the analysis had employed this DOAC as a treatment, thus, preventing us from detecting its real-size effect. Furthermore, statistically insignificant differences between types of DOACs could result from the fact that all DOAC types are substrates for the P-glycoprotein transporter, and they all interact with anticancer treatments to varying degrees [8,19,47]. Indeed, the administration of P-glycoprotein transporter inhibitors or CYP3A4 inhibitors, such as ketoconazole in cancer treatment, can enhance the exposure to DOACs, predisposing patients to a higher incidence of hemorrhagic events [47,48]. This is particularly true for apixaban and rivaroxaban, as they are mainly metabolized via CYP3A4 using a P-glycoprotein transporter mechanism [48]. On the other hand, strong P-glycoprotein transporter and CYP3A4 enhancers, such as rifampin, are able to decrease the bioavailability of apixaban and rivaroxaban by half, thus, decreasing the efficacy of these drugs [49]. In particular, for active prostate cancer, no pharmacological interferences with DOACS and docetaxel have been reported [50,51,52]. On the contrary, the association with hormone therapy is not advisable due to the interactions with both the cytochrome CYP3A4 and the P-glycoprotein transport system [53].

## 5. Limitations 

This meta-analysis has some significant limitations that need to be addressed. First, some of the included studies in the present research were retrospective, given the low number of prospective-randomized studies on the topic published in the literature. Indeed, oncologic patients are often excluded from prospective studies as their life expectancy is low. Second, the contributions of the single studies to the analysis were not homogenous in terms of the number of patients. Third, in the analysis of some outcomes, high heterogeneity was found. Fourth, it was not possible to draw final conclusions on the safety and efficacy of different types of DOACs because of the low number of single drugs used in the DOAC group compared to the VKA group. Finally, some of the studies included had explicitly focused on ischemic or hemorrhagic stroke, which could have determined an under- or over-estimation of the overall thromboembolic and bleeding incidence rates.

## 6. Conclusions 

DOACs showed better outcomes in terms of safety and efficacy than VKAs in patients with active and remote cancer, concomitantly affected by AF. DOACs showed a reduction in stroke, myocardial infarction, VTE, and major bleeding compared to VKAs. No statistically significant difference was found within the DOAC group. Further research on the topic is warranted.

## Figures and Tables

**Figure 1 jcm-11-05712-f001:**
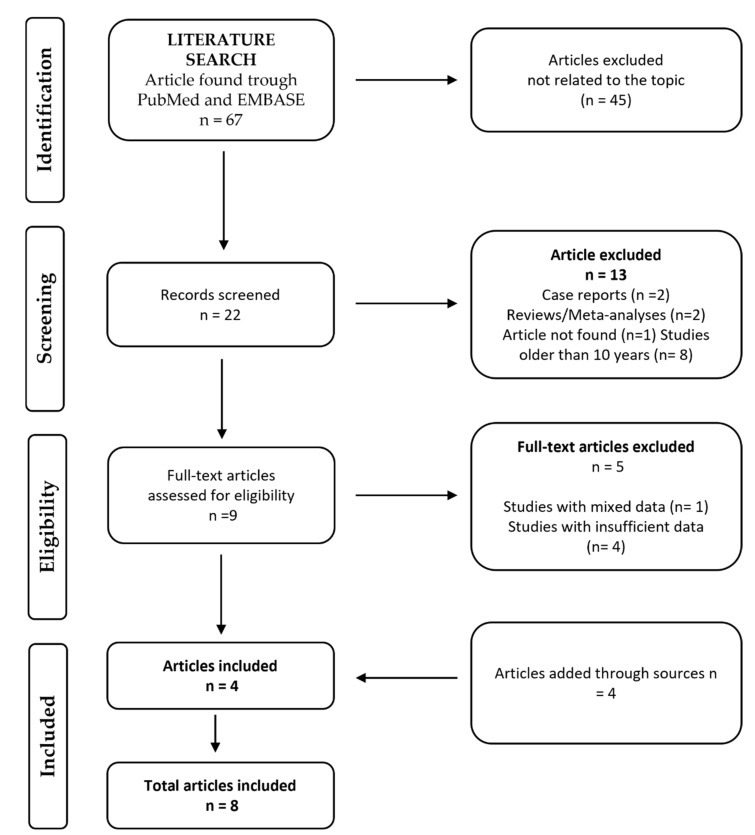
PRISMA flow chart of the selection process.

**Figure 2 jcm-11-05712-f002:**
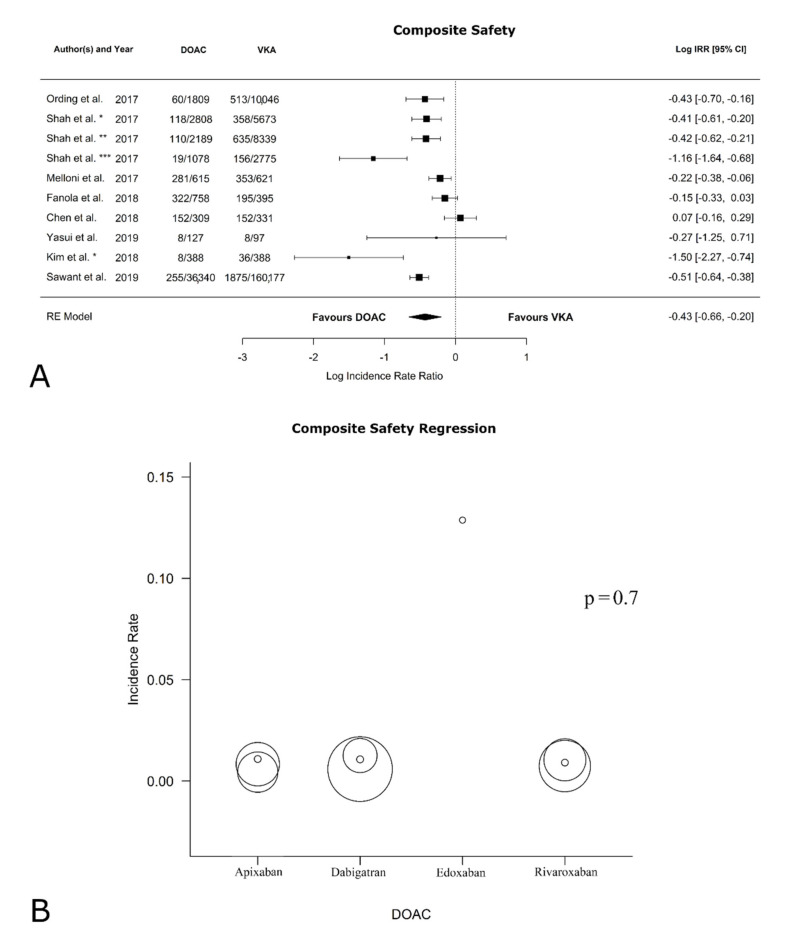
(**A**). Safety outcome composite. Forest plot of the IRR in VKAs versus DOACs. (**B**). Composite safety regression * rivaroxaban, ** dabigatran, *** apixaban [19,26,27,28,29,30,31,32].

**Figure 3 jcm-11-05712-f003:**
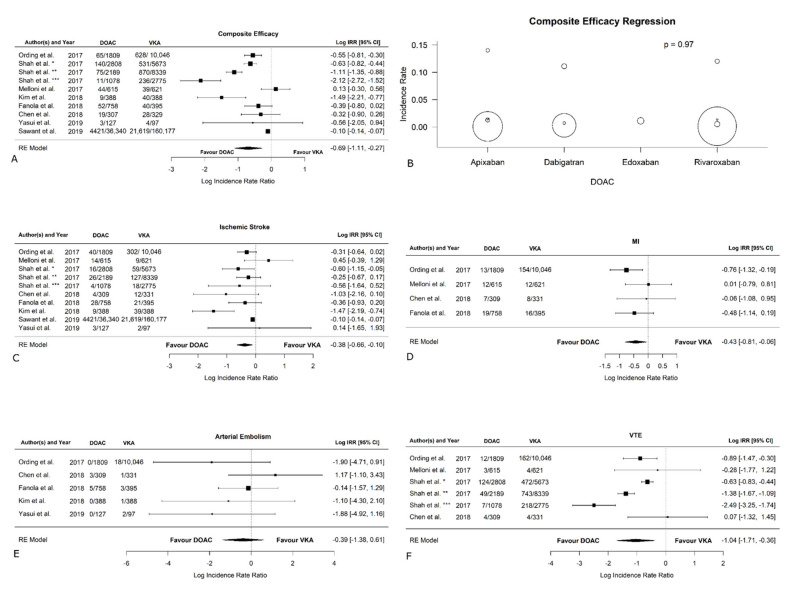
Efficacy outcomes in VKAs versus DOACs. (**A**). Composite efficacy (**B**). Composite efficacy regression. (**C**). Ischemic stroke. (**D**). Myocardial infarction. (**E**). Arterial embolism (**F**). Venous thromboembolic events. * rivaroxaban, ** dabigatran, *** apixaban [19,26,27,28,29,30,31,32].

**Figure 4 jcm-11-05712-f004:**
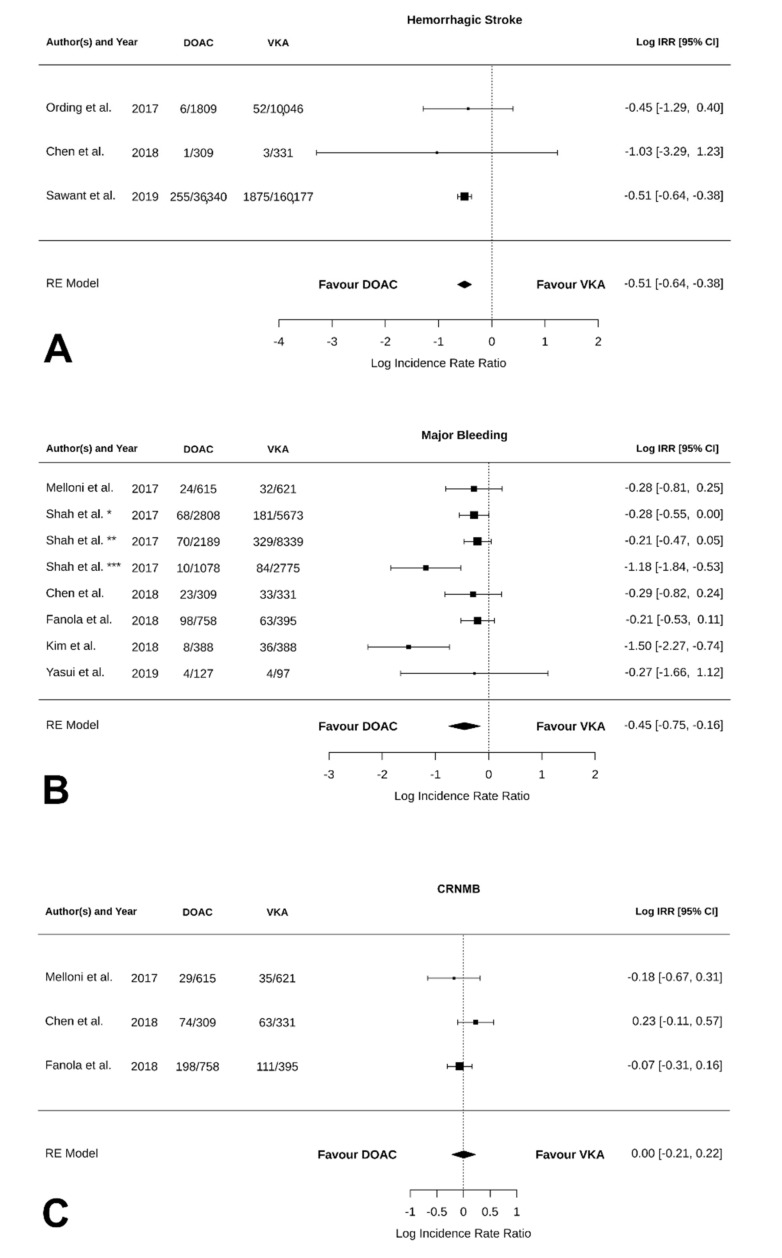
Forest plot of the IRR for safety outcomes in VKAs versus DOACs. (**A**). Hemorrhagic stroke. (**B**). Major bleedings. (**C**). Clinically relevant non-major bleeding. * rivaroxaban, ** dabigatran, *** apixaban [19,27,29,30,31,32].

**Table 1 jcm-11-05712-t001:** Characteristics of the studies and overall population.

Author	Year	Study Design	No. of Patients	Age	M/F	Cancer (Active/Remote)	Definition of Active/Remote Cancer		Anticoagulant		Type of VKA		Type of DOAC			
									VKA	DOAC	W	O	A	R	D	E
Ording et al. [29]	2017	OS	11,855	-	-	Active	Active: diagnosed < 2 years before the index date (=redemption date of the first reimbursed prescription of anticoagulants)		10,046 (84.7)	1809 (15.3)	-	-	-	-	-	-
Shah et al. [30]	2017	RCS (PS-matched)	16,096	-	-	Active	Active: use of chemotherapy, radiation therapy, or cancer surgery within 6 months prior to the start of anticoagulation		10,021 (62.3)	6075 (37.7)	10,021 (100)	0 (0)	1078 (17.6)	2808 (46.2)	2189 (36.2)	0 (0)
Melloni et al. [28]	2017	Post-hoc analysis from ARISTOTLE Trial (RCT)	1236	74 (68–80) * 75 (69–80) †	126/31 * 1004/75 †	Active: 157 (12.7) Remote: 1079 (87.3)	Active: malignancy other than basal or squamous cell skin cancer treated within the past 1 year Remote: medical disease history question malignancy other than basal or squamous cell skin cancer		621 (50.2)	615 (49.8)	621 (100)	0 (0)	615 (100)	0 (0)	0 (0)	0 (0)
Kim et al. [32]	2018	RCS (PS-matched)	776	-	-	Active	Newly diagnosed		388 (50)	388 (50)	388 (100)	0 (0)	138 (35.6)	110 (28.3)	140 (36.1)	0 (0)
Fanola et al. [19]	2018	Post-hoc analysis from ENGAGE-AF TIMI 48 Trial (RCT)	1153	75 (68–79)	794/359	Active	Active: new or recurrent malignancies other than nonmelanoma localized skin cancer, benign tumors, and in situ precancerous lesions		395 (34.3)	758 (65.7)	395 (100)	0 (0)	0 (0)	0 (0)	0 (0)	758 (100)
Chen et al. [27]	2018	Post hoc analysis from ROCKET AF Trial (RCT)	640	77 (72–81)	423/217	Active: 50 (7.8) Remote: 590 (92.2)	Active: patients receiving cancer treatment with hormonal or chemotherapeutic agents Remote: history of any cancer other than benign, pre-cancer, skin (except melanoma), basal, and squamous		331 (52)	309 (48)	331 (100)	0 (0)	0 (0)	309 (100)	0 (0)	0 (0)
Yasui et al. [31]	2019	RCS	224	72.7 ± 7.1	196/28	Active	Evidence of neoplasm on imaging or ongoing cancer therapy		97 (43.3)	127 (56.7)	97 (100)	0 (0)	46 (36.2)	44 (34.6)	25 (19.7)	12 (9.4)
Sawant et al. [26]	2019	ROS	196,517	76 ± 10	192,787/3730	Active	NS		160,177 (81.5)	36,340 (18.5)	160,177 (100)	0 (0)	9495 (4.8)	11,877 (6)	14,968 (7.6)	0 (0)
**Characteristics of the Population**																
**Author**			**Age** (**Average)**	**Gender**		**HTN**	**Diabetes**	**CHF**	**Pulmonary Disease**	**Renal Disease**	**Liver Disease**		**Metastasis**		**Hematological Malignancies**	
				**Female**	**Male**											
Ording et al. [29]	VKA n = 10,046 DOAC n = 1809		77 (70–83)	n = 4509 45%	n = 5537 55%	n = 6012 60%	n = 1434 14%	n = 901 9.0%	n = 2802 28%	n = 474 4.7%	n = 77 0.8%		n = 278 2.8%		n = 3.4	
Shah et al. [30]	VKA n = 10,021		75.4	40%												
	DOAC n = 6075		74.0													
Melloni et al. [28]	active cancer		74 (68–80)	n = 31 19.7%		n = 132 84.1%		n = 40 25.5%								
	remote cancer		75 (69–80)	n = 374 34.7%		n = 933 86.5%		n = 303 28.1%								
	VKA n = 621 DOAC n = 615															
Kim et al. [32]	VKA n = 1079		67.5	n = 387 31.5%		n = 804 74.5%	n = 403 37.3%	n = 295 27.3%		n = 127 11.8%			n = 153 14.2%			
	DOAC n = 572		74.2	n = 180 35.9%		n = 485 84.8%	n = 233 40.7%	n = 107 18.7%		n = 35 6.1%			n = 73 12.7%			
Fanola et al. [19]	VKA n = 395 DOAC n = 758		75		n = 794 68.9%	n = 1091 94.6%	n = 445 38.6%	n = 594 51.5%								
Chen et al. [33]	VKA n = 331 DOAK n = 309		77 (72–81)	n = 217 34%		n = 574 90%	n = 286 45%	n = 338 53%	111 17%				n = 4 <0.1%		n = 33 5.2%	
Yasui et al. [31]	VKA n = 97 DOAC n = 127		72.7 (±7.1)	n = 28 12.5%									n = 48 22.2%		n = 7 3.1%	
Sawant et al. [26]	VKA n = 160,177 DOAC n = 36,340		76 (±10)		98.1%	91.1%	57.0%	38.7%								

The studies are shown in order of year of publication. Values are expressed as mean ± standard deviation, median (interquartile range), or number (%). Abbreviations: A = Apixaban, D = Dabigatran, E = Edoxaban, NS = Not Specified, O = Others, OS = Observational Study, PS = Propensity Score, R = Rivaroxaban, RCS = Retrospective Cohort Study, RCT = Randomized Controlled Trial, ROS = Retrospective Observational Study, VKA = Vitamin K Antagonist, W = Warfarin. * Active cancer, † Remote cancer.

**Table 2 jcm-11-05712-t002:** Outcomes.

Author	Year	Ischemic Stroke		Myocardial Infarction		Venous Thromboembolism		Major Bleeding		Major or CRNM Bleeding		Any Bleeding (Major, CRNM, Minor)		Hemorrhagic Stroke		All-Cause Death	
		VKA	DOAC	VKA	DOAC	VKA	DOAC	VKA	DOAC	VKA	DOAC	VKA	DOAC	VKA	DOAC	VKA	DOAC
Ording et al. [29]	2017	1426 (14.2)	188 (10.4)	739 (7.4)	65 (3.6)	527 (5.2)	30 (1.7)	-	-	-	-	-	-	229 (2.3)	10 (0.6)	-	-
Shah et al. [30]	2017	59 (1.0) * 127 (1.5) † 18 (0.6) ‡	46 (0.8)	-	-	472 (8.3) * 743 (8.9) † 218 (7.9) ‡	180 (3.0)	-	-	2245 (39.6) * 3273 (39.2) † 551 (19.9) ‡	148 (2.4)	-	-	-	-	-	-
Melloni et al. [28]	2017	9 (0.8)	14 (1.3)	12 (1.1)	12 (1.1)	4 (0.4)	3 (0.3)	-	-	67 (6.9)	53 (5.5)	245 (32.2)	204 (26.5)	9 (0.9)	0 (0)	42 (3.6)	54 (4.7)
Kim et al. [32]	2018	39 (5.5)	9 (1.3)	-	-	-	-	8 (1.2)	-	-	-	-	-	-	-	93 (13.3)	41 (6.1)
Fanola et al. [19]	2018	21 (2.1)	28 (3.7)	16 (1.6)	19 (2.5)	-	-	63 (8.2)	98 (12.9)	174 (27.9)	296 (39.1)	195 (33.8)	322 (42.5)	-	-	120 (11.5)	241 (31.8)
Chen et al. [27]	2018	12 (2.0)	4 (0.7)	7 (1.2)	8 (1.4)	4 (0.6)	4 (0.7)	33 (6.4)	23 (4.7)	96 (21.6)	97 (23.6)	152 (40.8)	152 (46.6)	3 (0.6)	1 (0.2)	48 (8.0)	32 (5.4)
Yasui et al. [31]	2019	2 (2.1)	3 (2.4)	-	-	-	-	4 (4.1)	4 (3.1)	-	-	-	-	1 (1.0)	0 (0)	-	-
Sawant et al. [26]	2019	21,619 (13.5)	4421 (12.2)	-	-	-	-	-	-	-	-	-	-	1875 (1.2)	255 (0.7)	-	-

The studies are shown in order of year of publication. Values are expressed as number (%). Abbreviations: CRNM: Clinically Relevant Non-Major, DOAC = Direct Oral Anticoagulant, VKA = Vitamin K Antagonists. * Matched with rivaroxaban, † matched with dabigatran, ‡ matched with apixaban.

**Table 3 jcm-11-05712-t003:** Results of the analysis without Sawant et al. [26].

Active + Remote Cancer						
	IRR [95%CI]	*p*-Value IRR	I^2^ (%)	*p*-Value I^2^	Egger’s Intercept [95%CI]	*p*-Value Egger’s Test
**Thromboembolic events**	−0.77 [−1.22, −0.33]	0.0007	92.12	<0.0001	−0.64 [−1.30, 0.01]	0.01
**Ischemic stroke**	−0.45 [−0.77, −0.13]	0.006	52.59	0.05	−0.26 [−0.91, 0.39]	0.07
**Bleeding**	−0.43 [−0.70, −0.16]	0.002	89.52	<0.0001	0.03 [−0.32, 0.38]	0.01
**Hemorrhagic stroke**	−0.52 [−1.31, 0.28]	0.20	0.00	0.64	−0.1 [NA, NA] *	NA *
**Active Cancer Only**						
	**IRR [95%CI]**	***p*-Value IRR**	**I^2^ (%)**	***p*-Value I^2^**	**Egger’s Intercept [95%CI]**	***p*-Value Egger’s Test**
**Thromboembolic events**	−0.94 [−1.36, −0.52]	<0.0001	89.25	<0.0001	−0.61 [−1.14, −0.08]	0.00
**Ischemic stroke**	−0.51 [−0.81, −0.21]	0.0009	39.50	0.14	−0.18 [−0.69, 0.32]	0.03
**Bleeding**	−0.50 [−0.79, −0.21]	0.0008	84.76	0.0003	−0.12 [−0.50, 0.26]	0.01
**Hemorrhagic stroke** †	NA	NA	NA	NA	NA	NA

Abbreviations: CI = Confidence Interval, IRR = Incidence Rate Ratio, NA = Not Applicable. * It is not applicable because the R software does not calculate Egger’s test CI and *p*-value in only two studies. † Excluding Sawant et al., only one study had active-cancer-only patients, and thus, the analysis could not be carried out.

## Data Availability

Not applicable.

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
