# Peer review of "Superiority of Direct Oral Anticoagulants over Vitamin K Antagonists in Oncological Patients with Atrial Fibrillation: Analysis of Efficacy and Safety Outcomes"

_jcm, 2022, doi:10.3390/jcm11195712_

Round 1
Reviewer 1 Report
Overall, the paper provides a comprehensive overview of the topic.
With 7000 words it is relatively long and not easy to read. One solution might be to avoid redundancy in the results section between the text and figures.
The paper evaluates the data with and without Sawant and active vs non-active cancer. Perhaps it might be an option to report the events without Sawant separately / in summary? Removing Sawant reduces the analysis to 14% of the patients. In essence, Sawant dominates the entire paper. Removing Sawant entire and discussing the findings of this analysis vs Sawant could be considered.
An important discussion point are non-AF cancer patients treated with OACs, in other words: do AF patients differ from them? If not, then AF is unlikely and important factor in the OAC treatment of cancer patients.
Endpoints on the venous side are also included but not VTE patients.
Please review language.
How is prostate CA handled, especially when on antihormonal therapy, as active cancer??
Intro: AF causing cancer appears too strong. What needs discussing is that AF patients often undergo careful medical work up. In that context cancer is being diagnosed. There is also a distinction whether AF or the AF related treatment, i.e., the drugs administered, cause cancer. These events are extremely rare.
Lines 39-43: Some more stats is suggested
Lines 131-3: bullet points 5-7 are omissible
Lines 146: primary endpoint or outcome parameter(s). Efficacy, as this is the indication for treatment, should precede safety. Here it needs stating whether these are co-primary outcome parameters or a composite one.
Line 220: SD should accompany mean
Line 223: for NOACs the modified HAS-BLED as labile INR may not be assessed?
Line 345: no anticoagulant protects from bleeding events
Fig 5: after removal of Sawant almost 95% of the patients are as well. For CRNMB the numbers are very small relative to the others. Therefore, this adds more complexity to the review.
Lines 422-4: This statement is strong and a little daring.
Lines 477-81: Consider rephrasing this sentence.
Lines 507-13: Theoretically this could be correct. Given the many confounding factors hesitancy about the anti-neoplastic effect of FXa inhibitors may be appropriate.
Lines 554-9: The Sawant issues disappeared.
Author Response
Reviewer 1
- Thanks for the review. Overall, the provides a comprehensive overview of the topic. With 7000 words it is relatively long and not easy to read. One solution might be to avoid redundancy in the results section between the text and figures.
- ANSWER: We agree with the reviewer, we reduced the text and we deleted the description of figure s 2 from the text. The description has been reported in the figure 2 legends as it has been suggested. Line 378-392-404-534-467-499
- The paper evaluate-s the data with and without Sawant and active vs non-active cancer. Perhaps it might be an option to report the events without Sawant separately / in summary? Removing Sawant reduces the analysis to 14% of the patients. In essence, Sawant dominates the entire paper. Removing Sawant entire and discussing the findings of this analysis vs Sawant could be considered.
- ANSWER: We agree with the reviewer. However, we would respectfully outline that a sensitivity analysis was performed resulting in a lack of influence of Sawant’s paper on the final results. Therefore we respectfully ask this review to keep the Sawant work in order to maintain a more completeness of the information provided to a potential reader.
- An important discussion point are non-AF cancer patients treated with OACs, in other words: do AF patients differ from them? If not, then AF is unlikely and important factor in the OAC treatment of cancer patients.
- ANSWER: We agree with the reviewer. Unfortunately, this analysis could not be carried out because such a data was not provided by all the most of the authors.
- Endpoints on the venous side are also included but not VTE patients. .
- ANSWER: We agree with the reviewer. We would like to respectfully underline to this reviewer the point was addressed in table3-401
- Please review language.
- ANSWER: We agree with this insightful suggestion. English has been reviewed by a native teacher.
- How is prostate CA handled, especially when on antihormonal therapy, as active cancer
- ANSWER: We agree with the reviewer. We added the strategy suggested by the guidelines EHRA 2021. Line 691
- Intro: AF causing cancer appears too strong. What needs discussing is that AF patients often undergo careful medical workup. In that context, cancer is being diagnosed. There is also a distinction whether AF or the AF-related treatment, i.e., the drugs administered, cause cancer. These events are extremely rare
- ANSWER: We agree with the reviewer. We deleted that AF and the drugs administrated can cause cancer and we had better explained that AF could be a marker of occult cancer (see discussion p. line 543 -553)
- Lines 39-43: Some more stats is suggested
ANSWER: We agree with the reviewer. We improved the statistical analysis description in this section.line 39-41
Changes: Lines 131-3: bullet points 5-7 are omissible
-ANSWER: We agree with the reviewer. These bullet points have been omissed.line 157
- Lines 146: primary endpoint or outcome parameter(s). Efficacy, as this is the indication for treatment, should precede safety. Here it needs stating whether these are co-primary outcome parameters or a composite one.
-ANSWER: We agree with this reviewer and accordingly, we gathered main outcomes in the same figures. Starting with efficacy as requested by this reviewer. Furthermore, since we believe that the suggestion including also the outcome as composite endpoints we added a further figure reporting composite safety and composite efficacy respectively
Changes: As requested, we reported safety first. Furthermore, the endpoints have been better clarified in the figures 3 page 393 .The previous figure2 line 375 amd figure 3 line 393have been deleted- Composite line182
- Line 220: SD should accompany mean
- ANSWER: We apologise whether it is not clear that age was expressed as median and interquartile range since it was normally distributed. This was specified in the statistical section.
Chages: Added in statistical section.line 195
-Line 223: for NOACs the modified HAS-BLED as labile INR may not be assessed?
ANSWER: We thank the reviewer for the question. We assessed Hypertension, Abnormal liver/renal function, Stroke, Bleeding history or predisposition, Labile international normalized ratio, Elderly, Drugs/alcohol concomitantly (HAS-BLED)
- Line 345: no anticoagulant protects from bleeding events.
ANSWER: We agree with the reviewer. Accordingly, with this comment, we deleted the phrase “This suggests that DOACs are superior to VKAs in protecting from bleeding events”. And we had better clarified DOACs were superior to VKAs in terms of the risk of major bleeding.linepag 298
- Fig 5: after removal of Sawant almost 95% of the patients are as well. For CRNMB the numbers are very small relative to the others. Therefore, this adds more complexity to the review.
ANSWER: We agree with the reviewer. Thank you, the analysis took into account the small number of patients
- Lines 422-4: This statement is strong and a little daring.
ANSWER: We agree with the reviewer. Accordingly, with this comment, we changed this concept and we deleted that statement.line 586
- Lines 477-81: Consider rephrasing this sentence
ANSWER: We agree with the reviewer. Accordingly, with this comment, we rephrased and we clarified this sentence.linepag 588
- Lines 507-13: Theoretically this could be correct. Given the many, confounding factors hesitancy about the anti-neoplastic effect of FXa inhibitors may be appropriate.
ANSWER: We agree with the reviewer. This was specified as requested line 664
- Lines 554-9: The Sawant issues disappeared.
ANSWER: We agree with the reviewer. The results have been confirmed after sensitive analyses also.

Reviewer 2 Report
The superiority of DOAC over VitK antagonists in oncological patients with AF: analysis of efficacy and safety/reviewer comments
This meta-analysis by Parrini et al is important study. Although not novel by concept, its large number of patients and the appropriate statistical analyses are its main points of strength, and its conclusion of DOAC superiority over VitK antagonists is based on solid evidence and well proven. The statistical split of all studies with and without Sawant is appropriate. Few minor comments: 1. In the Introduction, I would elaborate on the hypercoagulative state which is intrinsic to cancer patients, explaining why anti-coagulant therapy is so important... Also, in the discussion, the authors only speak and mention of Trousseau syndrome, but is this the ONLY relation of malignancy to hypercoagulability? Probably not, and here the authors need to explain more about this relationship. 2. In the Results section, the authors need to add a new table summarizing the patients' characteristics between the group treated via DOACs and that treated by VitK antagonists. Currently, only age and mean CHADS are mentioned but I think it is important to know many other parameters as well, so to be able to see that the groups are quite similar. The other parameters to be included are age, gender, HTN, diabetes, presence of CHF, pulmonary and renal disease, liver disease/cirrhosis, etc...also yes/no presence of metastases and percent of hematological malignancies. Notably, the above table should be split into all studies with and without Sawant et al due to the different num of patients included and the fact that in his study propensity score matching was used.Author Response
Reviewer 2
This meta-analysis by Parrini et al is important study. Although not novel by concept, its large number of patients and the appropriate statistical analyses are its main points of strength, and its conclusion of DOAC superiority over VitK antagonists is based on solid evidence and well proven. The statistical split of all studies with and without Sawant is appropriate.
Few minor comments:
- In the Introduction, I would elaborate on the hypercoagulative state which is intrinsic to cancer patients, explaining why anti-coagulant therapy is so important...
ANSWER: We thank the reviewer for this insightful comment. Accordingly, we improved this statement adding that “Moreover, it is noteworthy that cancer patients with AF have not only the AF-related thromboembolic risk but also the intrinsic risk due to hypercoagulability and the prothrombotic state typically occurring in malignancy. It has been well assessed that tumor cells promote the activation of the coagulation system, secreting procoagulant factors and inflammatory cytokines. A complex interaction between tumor cells and blood and vascular cells has also been reported [6,7].”line 92
- Also, in the discussion, the authors only speak and mention of Trousseau syndrome, but is this the ONLY relation of malignancy to hypercoagulability? Probably not, and here the authors need to explain more about this relationship.
ANSWER: We agree with the reviewer. We changed it accordingly line 556
- In the Results section, the authors need to add a new table summarizing the patients' characteristics between the group treated via DOACs and that treated by VitK antagonists. Currently, only age and mean CHADS are mentioned but I think it is important to know many other parameters as well, so to be able to see that the groups are quite similar. The other parameters to be included are age, gender, HTN, diabetes, presence of CHF, pulmonary and renal disease, liver disease/cirrhosis, etc...also yes/no presence of metastases and percent of hematological malignancies.
ANSWER: We agree with the reviewer. Accordingly, with this comment, we added a new table (Tabel 1 A e linea 270) maintaining the distinction between VKAs and DOACs where it has been possible
- Notably, the above table should be split into all studies with and without Sawant et al due to the different num of patients included and the fact that in his study propensity score matching was used
ANSWER: We agree with the reviewer. As explained a sensitive analysis has been performed.
